# Peer J

# Ontogenetic scaling patterns and functional anatomy of the pelvic limb musculature in emus (*Dromaius novaehollandiae*)

Luis P. Lamas[1], Russell P. Main[2] and John R. Hutchinson[1]

[1] Structure and Motion Laboratory, Department of Comparative Biomedical Sciences, The Royal Veterinary College, Hatfield, United Kingdom
[2] Department of Basic Medical Sciences, College of Veterinary Medicine, Purdue University, West Lafayette, IN, USA

## ABSTRACT

Emus (*Dromaius novaehollandiae*) are exclusively terrestrial, bipedal and cursorial ratites with some similar biomechanical characteristics to humans. Their growth rates are impressive, as their body mass increases eighty-fold from hatching to adulthood whilst maintaining the same mode of locomotion throughout life. These ontogenetic characteristics stimulate biomechanical questions about the strategies that allow emus to cope with their rapid growth and locomotion, which can be partly addressed via scaling (allometric) analysis of morphology. In this study we have collected pelvic limb anatomical data (muscle architecture, tendon length, tendon mass and bone lengths) and calculated muscle physiological cross sectional area (PCSA) and average tendon cross sectional area from emus across three ontogenetic stages ($n = 17$, body masses from 3.6 to 42 kg). The data were analysed by reduced major axis regression to determine how these biomechanically relevant aspects of morphology scaled with body mass. Muscle mass and PCSA showed a marked trend towards positive allometry (26 and 27 out of 34 muscles respectively) and fascicle length showed a more mixed scaling pattern. The long tendons of the main digital flexors scaled with positive allometry for all characteristics whilst other tendons demonstrated a less clear scaling pattern. Finally, the two longer bones of the limb (tibiotarsus and tarsometatarsus) also exhibited positive allometry for length, and two others (femur and first phalanx of digit III) had trends towards isometry. These results indicate that emus experience a relative increase in their muscle force-generating capacities, as well as potentially increasing the force-sustaining capacities of their tendons, as they grow. Furthermore, we have clarified anatomical descriptions and provided illustrations of the pelvic limb muscle–tendon units in emus.

Corresponding author
Luis P. Lamas, llamas@rvc.ac.uk

## INTRODUCTION

Scaling studies (relating animal body mass to other biological parameters) have broadly elucidated locomotor adaptations across a wide range of body sizes. These studies have also described important size-related biomechanical (*Alexander et al., 1979*; *Bertram & Biewener, 1990*; *Biewener, 1982*; *Gatesy & Biewener, 1991*; *LaBarbera, 1989*; *Maloiy et al., 1979*; *McMahon, 1975*) and metabolic (*Gillooly et al., 2001*; *Hemmingsen, 1960*; *Hokkanen, 1986*; *Kleiber, 1932*; *Schmidt-Nielsen, 1984*; *Taylor et al., 1981*) constraints across species. Intraspecific scaling studies are less common (*Allen et al., 2010*; *Allen et al., 2014*; *Carrier & Leon, 1990*; *Carrier, 1983*; *Dial & Jackson, 2011*; *Main & Biewener, 2007*; *Miller et al., 2008*; *Picasso, 2012*; *Smith & Wilson, 2013*; *Young, 2009*; *Picasso, 2014*). These ontogenetic approaches yield valuable insights into musculoskeletal adaptations to growth and potentially to identify size-related constraints on mechanical function within a species. Furthermore, studies of species where the mode of locomotion and basic anatomy remains similar during development contribute to the understanding of strategies and trade-offs that occur during growth. Such information can, for example, be used to comprehend developmental abnormalities and study intervention strategies to manage them.

Ratites are large flightless birds with cursorial morphology (e.g., *Smith, Jespers & Wilson, 2010*; *Smith & Wilson, 2013*) that makes them attractive subjects for studies of terrestrial locomotion and bipedalism (*Abourachid, 2000*). Certain characteristics make emus (*Dromaius novaehollandiae*) particularly useful: they have some anatomical and functional similarities to other bipedal animals, including purportedly humans (*Goetz et al., 2008*). Compared to ostriches, they are generally easier to handle and train in experimental settings due to their smaller size and calmer temperament. Finally, their growth rate is impressive, as they multiply their body weight ∼80 times in the first 18 months of life (*Minnaar & Minnaar, 1997*) whilst maintaining the same cursorial mode of locomotion. Despite this interest, there are still some discrepancies in published anatomical descriptions and depictions of the pelvic limb musculature of emus (*Haughton, 1867*; *Patak & Baldwin, 1998*; *Vanden Berge & Zweers, 1993*), and clear visual anatomical aids are lacking in the literature.

Some of the biomechanical changes in the hindlimb occurring during the growth in emus have been described. *Main & Biewener (2007)* measured the skeletal strain patterns on the surfaces of the femur and the tibiotarsus (TBT) in running emus, demonstrating a significant increase in the magnitude of cranial and caudal femoral and caudal tibiotarsal strains during ontogeny, despite the enlargement and strengthening of those bones via positive allometric scaling of the second moment of area. Muscles have been shown to influence the strain patterns of bones (*Yoshikawa et al., 1994*), and although other factors are likely to be involved in the changes in peak bone strains reported across ontogeny (*Main & Biewener, 2007*), allometric scaling of the musculature could also play a role in these differences in bone tissue loading. The strains induced by muscle contraction will be proportional to the muscle forces acting on the bone; therefore by estimating muscle forces (e.g., maximal force capacity based upon anatomy), associations between these two findings would be possible.

In order to build on already available biomechanical data for emus (*Goetz et al., 2008*; *Main & Biewener, 2007*), we aim here to quantify the ontogenetic scaling patterns of four pelvic limb bones, pelvic limb muscles and their tendons and in the process describe and compare the functional and descriptive anatomy of the pelvic limb musculature of emus. We use regression analysis to determine the relationship of muscle architectural properties with body mass in an ontogenetic series of emus, and then examine the implications of these findings for the locomotor ontogeny of emus, other ratites, as well as extinct theropod dinosaurs.

## MATERIALS AND METHODS

### Animal subjects and care: UK group

We dissected 17 emus for this study, obtained from our ongoing research examining emu ontogenetic biomechanics (conducted with ethical approval under a UK Home Office license). The emus were divided in three groups of animals according to their age: Group 1: Five individuals at 4–6 weeks old; Group 2: Six 24–28 weeks (6 months) old individuals; and Group 3: Six 64–68 weeks (16 months) old individuals. All birds had been used as experimental animals and kept in a small pen (7 × 7 m) for the first six weeks of life, after which they were moved to an outdoor larger enclosure with grass footing (40 m × 15 m) until they were six months old; after this they were moved to a large (1.6 hectares) grass field (maximal animal density at one time was 8 birds/ha). The birds were all born in three consecutive yearly breeding seasons. Only the birds in Group 3 were from the same breeding season but not necessarily the same progenitors; birds from the other two Groups were from two different seasons.

All animals were hatched at a commercial breeding farm in the UK and raised from four weeks of age at the Royal Veterinary College. They were fed a commercial ostrich pelleted diet supplemented with grass, and from six weeks of age were kept with free access to commercial food and grass. At 24 weeks, their diet changed from an ostrich grower diet to adult ostrich pelleted food (Dodson and Horrel Ltd., Kettering, Northamptonshire, UK). There were no restrictions or enforcements on the animals' regular exercise regime, and all animals were allowed the same area and conditions to exercise during their development. All animals were euthanized after other experimental procedures were completed, by lethal intravenous injection of a barbiturate following induction of deep terminal general anaesthesia by intramuscular injection of ketamine and xylazine. Carcasses were kept frozen in a −20 °C freezer for up to 2 years before dissection. Thawing was allowed at variable ambient temperatures and for variable amounts of time depending on the size of the animal, and dissection started no longer than 4 days after removal from the freezer. All dissections were performed within a six week period and led by the same individual (L.P.L.).

### USA group of emus

Unpublished raw data of muscle masses from a different group of 29 emus (0.74–51.7 kg body mass) used for similar purposes as those described for the UK group were also included in this study. This group was bred and reared in the USA (Concord Field Station,

Harvard University) under the care of another investigator (R.P.M.) who led all dissections for this group. The size and age composition for this group was more heterogeneous, and only body masses and muscle masses were available for analysis. Because the purpose of the dissections in the group was not a systematic ontogenetic musculoskeletal scaling study, the number of muscles dissected per animal varied.

## Bone measurements

Maximal interarticular lengths of the femur, tibiotarsus (TBT), tarsometatarsus (TMT) and first phalanx of the middle (third) digit were measured using an ordinary flexible measuring tape ($\pm 1$ mm) once they were cleared of all soft tissues.

## Myology and muscle architecture

We identified muscles of emus using four separate literature sources (*Haughton, 1867*; *Patak & Baldwin, 1998*; *Smith et al., 2007*; *Vanden Berge & Zweers, 1993*); when our observations differed from these, we described the anatomical landmarks and attachments in detail according to our observations. General main actions of the muscle were defined based on these publications and confirmed by identifying the muscle attachments and paths and then mimicking the muscle action by applying tension on the muscle during dissection. We used additional reference to a biomechanical model of an ostrich (*Hutchinson et al., 2014*) to refine the three-dimensional actions of the hip muscles, as those actions are difficult to accurately ascertain from visual inspection and manipulation. Table 1 shows our simplified description of the anatomy, abbreviations used throughout this study, and inferred muscle actions. Figures 1–3 show schematic anatomical representations of the muscle anatomy.

To avoid freeze drying of the carcasses, we ensured all animals were frozen soon after euthanasia and kept in sealed bags, and were not thawed and refrozen before dissection. The carcasses showed minimal autolysis and therefore an easier and better dissection during which muscle actions could be approximated without damaging their structure and attachments.

Dissection of the right pelvic limb muscles was performed in all specimens apart from the first two subjects in the 4–6 week old group, in which the muscles of the left limb were dissected first to standardise the technique. Measurements taken from the muscles of the left limb were not used (avoiding duplication of information), with the exception of when there were unidentified/damaged muscles from the right limb of the same specimen, in order to create a complete set of muscles for each specimen.

After identification of each muscle, we performed complete dissection and removal of it by transection at its origin and insertion(s). Next, the muscle was laid flat on a table and we took four muscle architectural measurements in a standard protocol: muscle mass ($M_m$), fascicle length ($L_f$), muscle belly length and pennation angle ($\theta$). Muscle mass was measured on an electronic scale ($\pm 0.01$ g) after removal of tendons, fat and aponeuroses. Fascicle length was measured from at least five random sites within the muscle belly using digital callipers ($\pm 0.1$ mm). Muscle belly length was measured as the length ($\pm 1$mm) from the origin of the most proximal muscle fascicles to the insertion of the most distal fascicles

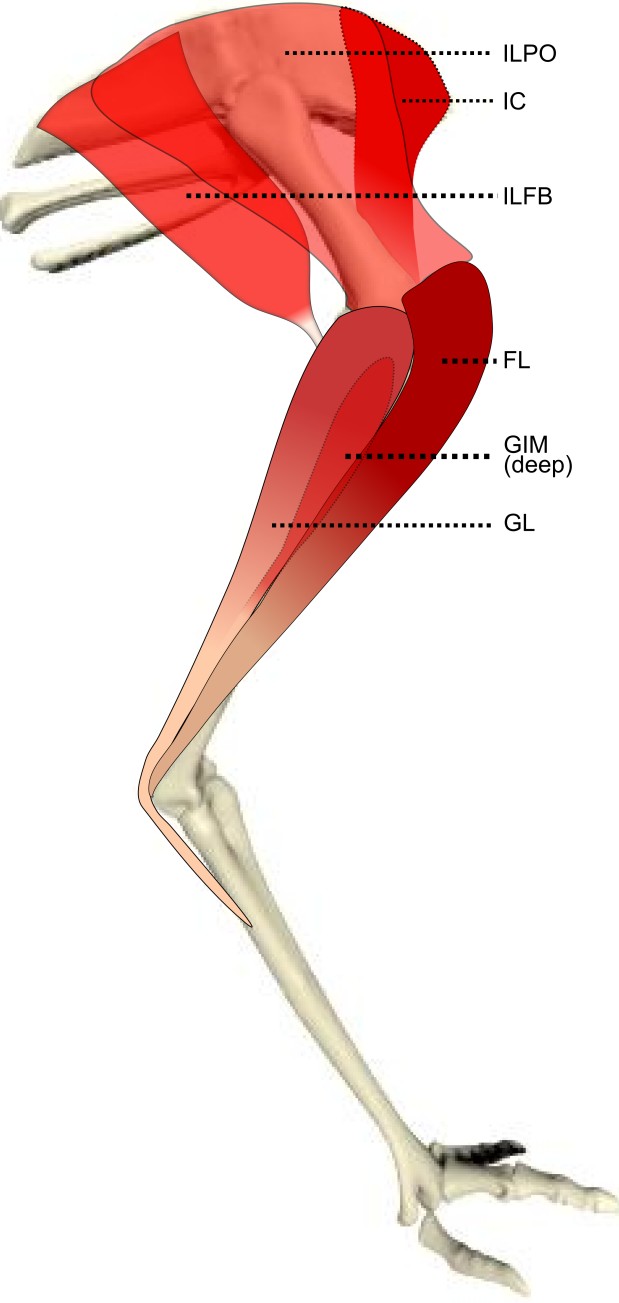

**Figure 1 Schematic anatomical representation of Emu pelvic limb anatomy.** Schematic anatomical representation of the most superficial layer of muscles, in lateral view, of the pelvic limb of an adult emu.

into the distal tendon or aponeurosis. The pennation angle was measured at least five times using a goniometer ($\pm 5°$) with the mean of the latter measurements taken as the pennation angle for the muscle. The repeated measurements were taken from multiple cuts into the muscle to expose different anatomical orientations of the fascicles with the same muscle. This methodology minimises the differences that may be seen across an individual muscle

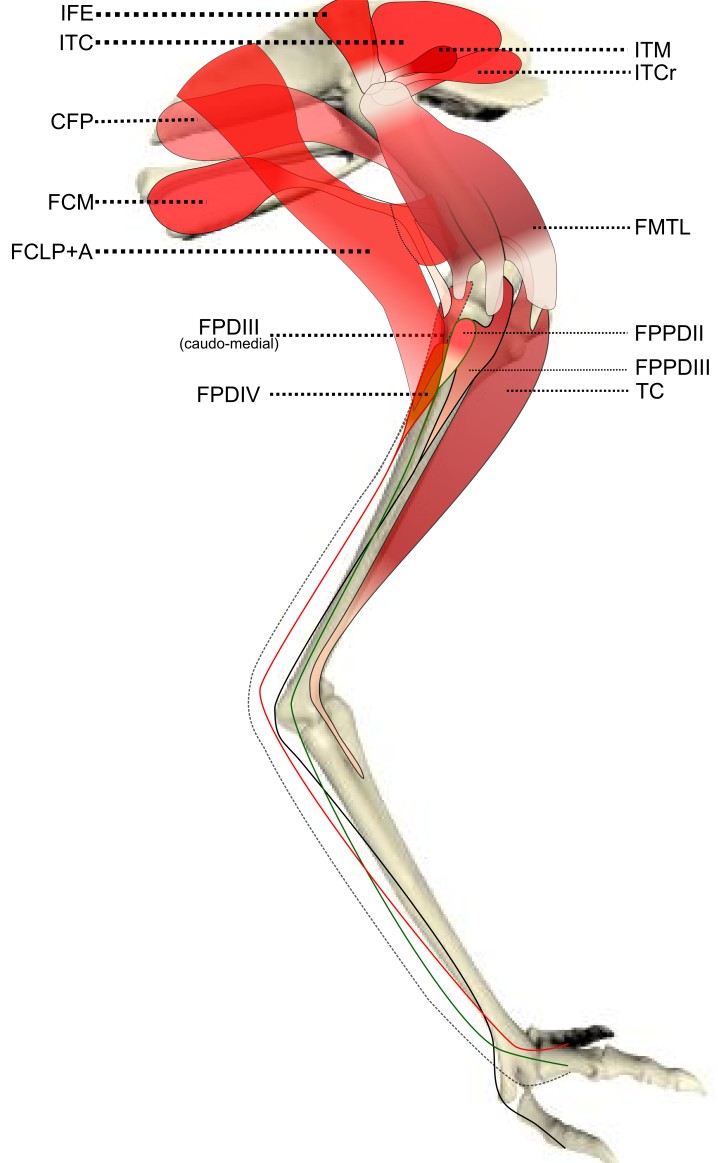

**Figure 2 Schematic anatomical representation of Emu pelvic limb anatomy.** Schematic anatomical representation of the intermediate layer of muscles, from a lateral view, of the pelvic limb of an adult emu.

and ensures mean values used for further calculations are representative of the overall architecture of the muscle. We calculated total limb muscle mass by adding the individual masses of the muscle bellies. Our approach was straightforward for most muscles, apart from three smaller muscles of the limb: IFI, ISF and FPPDII (Table 1), where minor dissection mistakes might have impaired estimates of their masses and architectural properties.

Muscle volume was calculated by dividing muscle mass by estimated muscle density of vertebrates (1.06 g cm$^{-3}$; *Brown et al., 2003*; *Hutchinson et al., 2014*; *Mendez & Keys, 1960*). From these data we calculated physiological cross-sectional area (PCSA) for each muscle

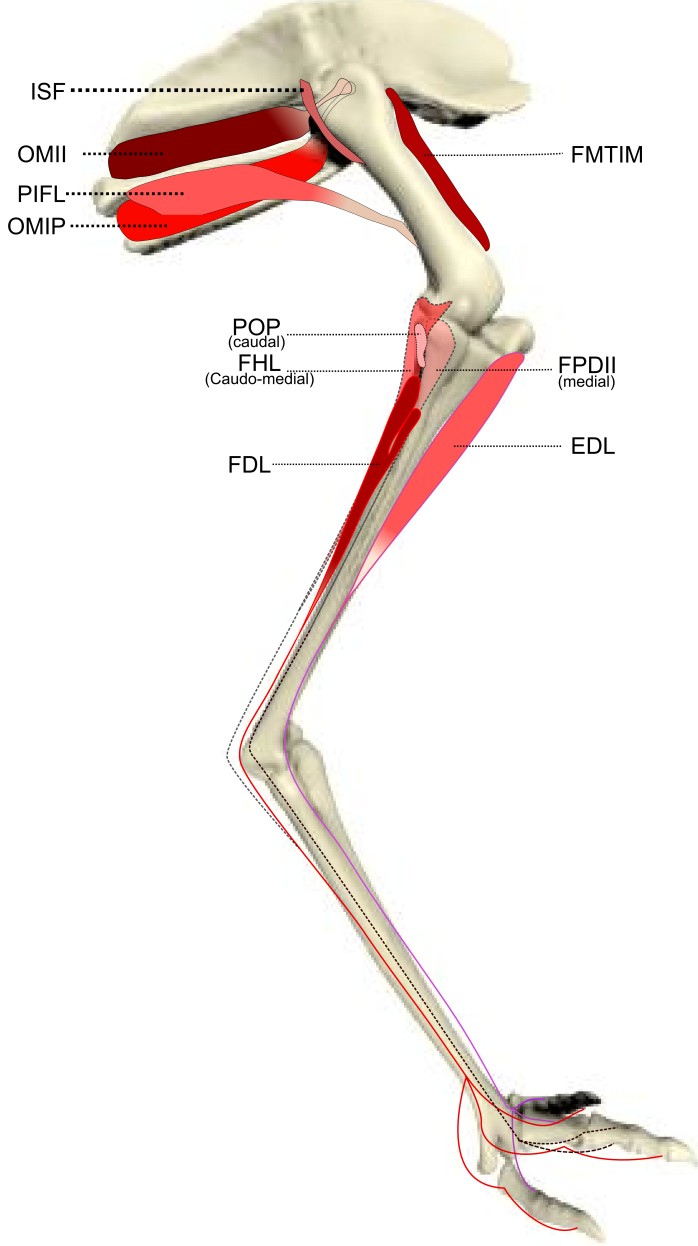

**Figure 3 Schematic anatomical representation of Emu pelvic limb anatomy.** Schematic anatomical representation of the deeper layer of muscles, from a lateral view, of the pelvic limb of an adult emu.

via the standard formula (*Powell et al., 1984*; *Sacks & Roy, 1982*):

$$\text{PCSA} = \left( \frac{Vmusc}{Lfasc} \right) \cos\theta. \tag{1}$$

When a tendon was present it was dissected down to its insertion onto the bone together with the muscle. The tendon was then transected at the musculotendinous junction when

**Table 1 Pelvic limb muscles of emus and their apparent actions.**

| Muscle | Abbreviation | Origin | Insertion | Action |
|---|---|---|---|---|
| M. iliotibialis cranialis | IC | Dorsal edge of preacetabular ilium | Insertion on the medial aspect of the proximal tibiotarsus | Main: Hip flexion; knee extension/flexion<br>Other: Hip medial rotation, adduction |
| M. iliotibialis lateralis (cranial and caudal portions) | ILPO | Lateral edge of acetabular ala | Craniolateral proximal tibiotarsus (cranial and lateral cristae cnemiales) via aponeurosis (combined with FMTL) | Main: Hip extension, ab-duction; knee extension<br>Other: Hip medial/lateral rotation |
| M. iliotrochantericus cranialis | ITCr | Cranial surface of preacetabular ilium | Lateral aspect of the femoral trochanteric crest (distal to IFE insertion) | Main: Hip flexion, medial rotation<br>Other: Hip abduction/adduction |
| M. iliotrochantericus medialis | ITM | Craniodorsal surface of preacetabular ilium | Lateral aspect of the femoral trochanteric crest (proximal to IFE insertion) | Main: Hip flexion, medial rotation<br>Other: Hip abduction/adduction |
| M. iliotrochantericus caudalis | ITC | Ala preacetabularis ilii: fossa iliaca dorsalis | Lateral aspect of the femoral trochanteric crest | Main: Hip flexion, medial rotation<br>Other: Hip abduction/adduction |
| M. iliofibularis | ILFB | Ala postacetabularis ilii: facies laterales | Proximal third of the corpus fibulae | Main: Knee flexion; hip extension<br>Other: Hip abduction |
| M. iliofemoralis externus | IFE | Crista iliaca dorsalis, dorsal to foramen acetabulum | Lateral side of femoral trochanteric crest (between ITC and ITM insertions) | Main: Hip flexion, abduction<br>Other: Hip medial/lateral rotation |
| M. iliofemoralis internus | IFI | Ventral preacetabular ilium | Medial side of proximal femoral shaft; tubercle | Main: Hip flexion, adduction<br>Other: Hip medial/lateral rotation |
| M. ischiofemoralis | ISF | Cranial margin of the foramen ilioischiadicum | Proximal caudal femur under origin of FMTL | Main: Hip abduction, lateral rotation<br>Other: Hip flexion/extension |
| M. caudofemoralis p. pelvica | CFP | Caudolateral ilium and ischium | Proximal caudomedial femur | Main: Hip extension<br>Other: Hip lateral rotation, abduction |
| M. flexor cruris lateralis pars pelvica | FCLP | Caudolateral corner of pelvis | Proximal craniomedial tibiotarsus | Main: Hip extension, abduction<br>Other: Medial rotation of hip and knee; knee flexion |
| M. flexor cruris lateralis pars accessoria | FCLA | By a raphe from the distal third of the FCLP | Caudomedial femoral shaft | Main: Hip extension, abduction<br>Other: Hip medial rotation |
| M. flexor cruris medialis | FCM | Caudolateral extremes of ischium and pubis | Via split cranial aponeurosis: on the caudal femoral shaft, and on the caudoproximal tibiotarsus, caudodistally to the insertion of the FCLP. | Main: Hip extension, abduction; knee flexion<br>Other: Hip medial rotation |
| M. puboischiofemoralis p. lateralis and p. medialis | PIFLM | Along the length of the lateral ischium | Via a thin tendinous insertion onto the caudal aspect of the femoral shaft | Main: Hip extension, abduction<br>Other: Hip lateral rotation |
| M. femorotibialis lateralis (Cranial, intermediate and caudal portions) | FMTL | Caudolateral surface of femoral shaft. With 3 fused parts: cranial, intermediate and caudal | Crista cnemalis of tibiotarsus via a thick patellar tendon (no ossified patella) with ILPO | Knee extension |

*(continued on next page)*

Table 1 (*continued*)

| Muscle | Abbreviation | Origin | Insertion | Action |
|---|---|---|---|---|
| M. femorotibialis intermedialis | FMTIM | Cranial surface of the proximal femoral shaft | Medial side of crista cnemalis cranialis of tibiotarsus | Knee extension |
| M. femorotibialis medialis | FMTM | 3 distinct heads originating from the medial surface of the femur, cranial and caudal portions on the proximal third and distal portion on the distal third | Proximo-medial extremity of tibiotarsus | Knee flexion, adduction |
| M. obturatorius medialis (Ilium—Ischium part) | OMII | Surface of fenestra ilioischium | Long tendon that passes through the foramen ilioischiadicum and inserts onto the lateral side of the femoral trochanteric crest | Main: Hip lateral rotation Other: Hip flexion, adduction |
| M. obturatorius medialis (Ischium—pubis part) | OMIP | Surface of fenestra ischiopubica | As OMII | Main: Hip lateral rotation Other: Hip flexion, adduction |
| M. ambiens | AMB | Cranial pubic rim (preacetabular process) | Two insertions on the medial knee ligaments, one tendinous and the other one fleshy | Main: Hip adduction; knee flexion Other: Hip medial rotation |
| M. gastrocnemius lateralis | GL | Lateral condyle of femur, aponeurosis of M. Iliotibialis and tendon from cranial fibula | Tendons fusing to form a thick fibrous calcaneal pad, onto caudal side of tarsometatarsus (Calcaneal scuttum) | Main: Ankle extension; knee flexion |
| M. gastrocnemius medialis | GM | Aponeurosis of M. Iliotibialis and facies gastrocnemialis, connecting to the medial surface of the proximal tibia | As GL | Main: Ankle extension; knee flexion |
| M. gastrocnemius Intermedius | GIM | Craniolateral femur, adjacent of the origin of FHL muscle | As GL and GIM | Main: Ankle extension; knee flexion |
| M. fibularis longus | FL | Proximal origin from medial distal patellar ligament and craniolaterally onto proximal tibiotarsus. | Two tendinous insertions: Plantar calcaneal scuttum and joining the tendon of FPDIII | Main: Ankle extension Other: Knee flexion; toe flexion via FPDIII tendon |
| M. tibialis cranialis c. tibiale and c. femorale | TC | 2 heads: A fleshy one onto the proximal cranial tibiotarsus, and via a thick tendon onto the cranial aspect of the lateral trochlear ridge of the femur | Cranial side of proximal tarsometatarsus | Main: Ankle flexion Other: Knee extension (femoral head) |
| M. popliteus | POP | Medial side of proximal fibula | Caudal side of proximal tibiotarsus | Main: Fibular rotation |
| M. flexor perforatus digiti II | FPDII | Via origin of FPDIII | Splits into 2 branches at level of proximal phalanx to insert on either side of middle phalanx, ventrally | Main: Digit II flexion Other: Ankle extension |
| M. flexor perforatus digiti III | FPDIII | 2 tendons: Cranial fibula and medial side of the medial condyle of the femur | Proximal phalanx, small portion fused to FPPDII tendon in some specimens, ventrally | Main: Digit III flexion Other: Ankle extension |

*(continued on next page)*

Table 1 (*continued*)

| Muscle | Abbreviation | Origin | Insertion | Action |
|---|---|---|---|---|
| *M. flexor perforans et perforatus digiti II* | FPPDII | Deep fibular tendon of GL muscle | Middle phalanx of digit II, ventrally | Main: Digit II flexion<br>Other: Ankle extension |
| *M. flexor perforans et perforatus digiti III* | FPPDIII | Lateral knee ligaments and FPDIV origin | Middle phalanx of digit III, ventrally | Main: Digit III flexion<br>Other: Ankle extension |
| *M. flexor perforatus digiti IV* | FPDIV | Superficial side of FPDIII origin | Proximal and middle phalanges of digit IV, ventrally | Main: Digit IV flexion<br>Other: Ankle extension |
| *M. flexor hallucis longus* | FHL | 2 heads: lateral and caudal aspects of distal femur near condyles | Fuses with FDL tendon | Main: Ankle extension; knee flexion |
| *M. flexor digitorum longus* | FDL | 2 heads: proximal tibiotarsus and distal third of fibula (3/4 of length) | Splits into 3 parts above MTP joint to insert onto the distal, ventral phalanx of each toe | Main: Digits II, III and IV flexion<br>Other: Ankle extension |
| *M. extensor digitorum longus* | EDL | Cranial proximal tibiotarsus | Dorsal surface of each phalanx | Main: Digits II, III and IV extension; ankle flexion |

a clear separation became apparent and stretched on a flat surface. We then measured lengths with a standard ruler or flexible measuring tape ($\pm 1$ mm), and tendon mass was also measured using the same instrumentation as for the muscles.

Tendon cross-sectional area (TCSA) was calculated using tendon length ($L_{ten}$); from muscle origin to bony insertion; and tendon mass ($M_{ten}$) as follows:

$$\text{TCSA} = \frac{M_{ten}}{1120\,L_{ten}} \tag{2}$$

where 1,120 kg m$^{-3}$ is assumed as the density of tendon (*Hutchinson et al., 2014*; *Ker, 1981*).

### Statistical analysis

Ontogenetic scaling relationships of (non-normalized) muscle properties were analysed using reduced major axis ("Model II") regression for $\log_{10}$ of each property vs $\log_{10}$ body mass using custom-designed R software code (*R Development Core Team, 2010*). A Shapiro–Wilk test was performed to assess normality of distribution of the residuals, and the *p* value for significance was set to $<0.05$. The inclusion criteria for data presented were: Datasets first had to have a *p* value $<0.05$ in the above described Shapiro–Wilk test. If this *p* was $>0.05$, the data were then tested for the presence of outliers (which were set at $\pm 2$ standard deviations [SD] from the mean) and outliers removed. The RMA linear regression was performed again using this dataset; and again, data were only presented if the *p* value for distribution of residuals was $<0.05$. Once the datasets were defined, $R^2$ correlation values and upper and lower bounds of the 95% confidence interval (CI) were calculated to assess the spread of data points around each regression line.

We used body mass (BM) as our independent variable and the target architectural parameter as our dependant variable. Overall, we followed a similar approach as that described by *Allen et al. (2010)* and *Allen et al. (2014)*.

Briefly, for two objects to be considered geometrically similar (and thus for an isometric scaling pattern to be inferred), areas should scale to the square product of lengths and volumes to the cube of lengths. Because mass is a volumetric property, the dependant variable is considered to scale isometrically if the mass of the structure scales with BM$^1$, areal properties (PCSA, TCSA) scale to BM$^{0.67}$ and lengths scale to BM$^{0.33}$, whereas angles and other non-dimensional variables should scale as BM$^0$. In order to obtain relative values to compare results from individuals of different size, muscle mass, PCSA and $F_{length}$ were normalized to body mass (BM) by dividing each value by the subject's BM, BM$^{0.67}$ and BM$^{0.33}$ respectively.

## RESULTS

We obtained 6,524 measurements of seven different muscle–tendon architectural parameters from 34 pelvic limb muscles and four pelvic limb bones in 17 emus from 3.6 to 42 kg of body mass. We found strong evidence for positive allometric scaling for many of these architectural parameters, as described below. To aid interpretation of our results, we have divided the muscles of the limb into proximal (those acting mostly on the

**Table 2 Regression analysis results for the lengths of the four limb bones.** The lower 95% boundary (>0.33) demonstrates positive allometry of the tibiotarsus and the tarsometatarsus but results are closer to isometry for the femur and first phalanx of digit III.

| Bone | Scaling exponent | Lower 95% CI | Upper 95% CI | $R^2$ |
|---|---|---|---|---|
| Femur | 0.38 | 0.34 | 0.42 | 0.96 |
| Tibiotarsus | 0.41 | 0.38 | 0.45 | 0.97 |
| Tarsometatarsus | 0.44 | 0.39 | 0.49 | 0.96 |
| First phalanx (Dig III) | 0.39 | 0.33 | 0.46 | 0.91 |

hip and knee joints) and distal (those acting on the ankle, foot and digits) groups and have used this division to compare trends between the two regions.

## Bone lengths

The lengths of the four bones scaled with moderate positive allometry (expected slope representing isometry would be 0.33). The femur had the least marked allometric exponent (0.38), whilst the tarsometatarsus had the greatest (0.44), the tibiotarsus had a slope value of 0.41 and for the 1st phalanx of the second digit (P1) the value was 0.39 (for full results see Table 2).

## Myology, architectural characteristics of muscles and functional interpretation

We classified a total of 34 muscles in Table 1. As noted by *Regnault, Pitsillides & Hutchinson (2014)*, there is no patellar ossification in the knee joint of emus, unlike ostriches and some other palaeognaths as well as most extant birds. Although muscle origins, insertions and paths were generally found to agree with previous publications (*Haughton, 1867*; *Patak & Baldwin, 1998*; *Vanden Berge & Zweers, 1993*) and hence detailed re-description is unnecessary, there were a few muscles for which we have found some differences worth noting, or for which we needed to use methodological simplifications:

M. iliotibialis lateralis pars postacetabularis (IL): The distal fusion and similar actions of both parts of the IL muscle (Fig. 1) meant that, in order to avoid dissection errors when finding the division between the cranial and caudal parts of the muscle, we measured and presented them together.

M. iliotrochantericus cranialis (ITCR): Although this was a clear, separate muscle in most specimens (Fig. 2), it was found to be fused with the ITM in two specimens of body mass ~20 kg, which is a common finding in birds (*Gangl et al., 2004*).

M. ischiofemoralis (ISF): This small muscle is difficult to detect and dissect, which is likely to have affected the accuracy of the data obtained from it (leading to lower $R^2$ values and wider 95% CI ranges). Its action is likely to involve fine motor control, proprioception and stabilisation of the hip joint, given its very small size. Some studies have considered this muscle to be absent (or fused with other muscles; e.g., CFP) in emus (*Haughton, 1867*; *Patak & Baldwin, 1998*), which would be unusual for any birds. The origin and insertion of the muscle that we label the ISF (Fig. 3) is best interpreted as a reduced—but still present—muscle, similar to that in ostriches (*Gangl et al., 2004*; *Zinoviev, 2006*).

M. caudofemoralis pars pelvica (CFP): We consider, contrary to other reports (*Haughton, 1867*; *Patak & Baldwin, 1998*), that this muscle is present in emus (Fig. 2). Prior studies classified this muscle as the "iliofemoralis" but we agree with the *Nomina Anatomica Avium* (*Vanden Berge & Zweers, 1993*) and other reports (*Gangl et al., 2004*; *Hutchinson, 2004a*; *Hutchinson et al., 2014*; *Zinoviev, 2006*) that it is present in ratites, related to a reduced portion of the large caudofemoralis muscle that is ancestrally present in tailed reptiles (*Gatesy, 1999*). There is no evidence of a caudalis part to the M. caudofemoralis in emus, unlike in ostriches (*Gangl et al., 2004*) and some other ratites, so this sub-division of the CFP is either fused to the CFP or lost.

M. ambiens (AMB): We found this muscle to have two insertions, previously unnoticed: a tendinous one onto the tibia and a fleshy one onto the distal femur. Unusual modifications of this muscle seem common in ratite birds (*Hutchinson et al., 2014*).

M. popliteus: This is a short, deeply positioned, fleshy muscle with multiple fibrous planes within it, originating on the caudolateral, proximal aspect of the tibiotarsus and inserting onto the medial side of the proximal fibula (Fig. 3). It is likely a stabiliser or pronator/supinator of the fibula, as in ostriches (*Fuss, 1996*), and may act as a proprioceptive or ligament-like structure.

## Normalized data for individual muscles

To allow relative comparisons between muscle measurements, we normalized data from only the 16 month old (Group 3, adult birds) emus. Data are only presented for adult emus so as not to influence the relative proportions due to ontogenetic allometry. The entire ontogenetic data set was analysed similarly with negligible differences found, indicating that the relative patterns seen between muscles for adults are present in very young birds as well. Data for muscle mass, fascicle length and PCSA are presented in Fig. 4. The largest relative muscles with regards to mass were three proximal (ILPO, ILFB and IC) and three distal muscles (GM, FL and GL). This order changes when muscles are ranked according to PCSA because parallel-fibred muscles tend to drop down the list, with the large ILPO being the only parallel-fibred muscle seen in the top 10 of a list that is otherwise dominated by distal muscles (FL, GM, GL and FPDIII). On the other hand, when fascicle length is compared, the three parts of the gastrocnemius (GIM, GM and GL) are the only distal muscles listed amongst the 10 muscles with the longest fascicles. The four muscles with the longest fascicles are the FCLP, IC, ILFB and ILPO.

## Limb muscle masses

Total mass values of the hindlimb musculature represented a mean of $13.4 \pm 3\%$ of BM, with the proximal limb musculature (PLM) representing $61 \pm 2\%$ of limb muscle mass and the distal limb muscles (DLM) accounting for the remaining $39 \pm 2\%$. However, if only values for the six largest birds (adults) are analysed, limb muscle mass accounts for $15 \pm 1\%$ of BM. The limb muscle mass is only $11 \pm 3\%$ of body mass in the five birds that were 4–6 weeks old.

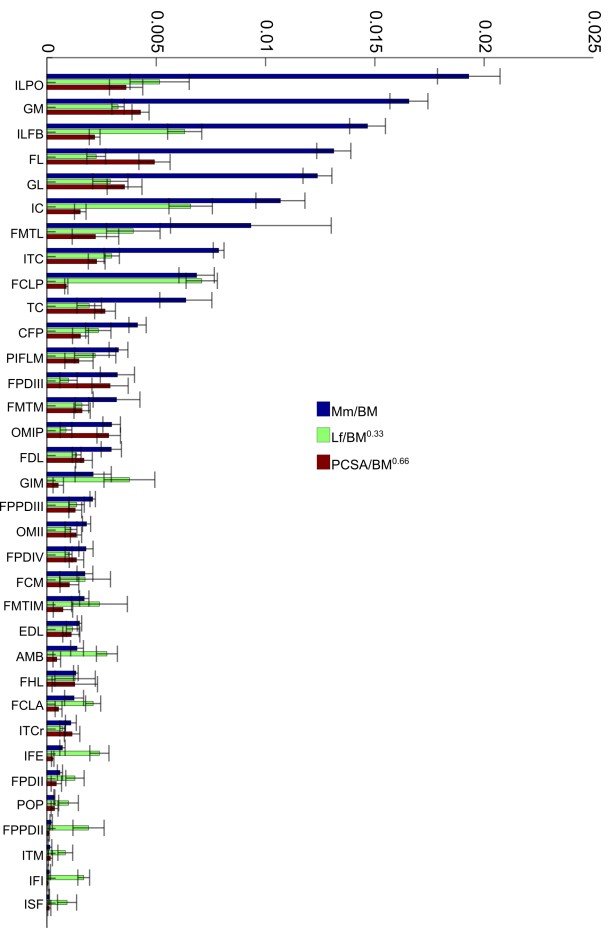

**Figure 4  Normalized data: 16 month old group only.** Normalized relative muscle parameters for individual muscles in emu pelvic limbs of the 16 month old birds only (Group 3; mean body mass 38.5 kg); mean values (error bars showing ±1 S.D.) are shown. Abbreviations for muscles are in Table 1. The key on the right side of the figure shows how muscle mass ($M_m$), physiological cross-sectional area (PCSA), and fascicle length ($L_f$) were normalized. $L_f$ values were adjusted to be 1/10 of the actual results in order to be of similar magnitude to the others. Muscles are organised from top to bottom in decreasing order of muscle mass.

## Scaling regression analysis

The slopes of the reduced major axis regression lines for muscle properties vs. body mass are shown in Tables 3 and 4, with $R^2$ and 95% CIs, as well as represented in Figs. 5A and 5B and 6. Ranges of the slope and amplitudes of the CIs referred to below are the upper and lower bounds of the 95% CIs for the regression slopes. Scaling exponents and CIs are presented in Table 3. Scaling exponents and lines representing isometry are plotted in Figs. 5A and 5B ($M_m$, $L_f$ and PCSA) and Fig. 7 ($M_{ten}$, $L_{ten}$ and TCSA). In summary, there was strong positive allometry of muscle mass and mild positive allometry or isometry of fascicle length, leading to a marked positive allometry of PCSA.

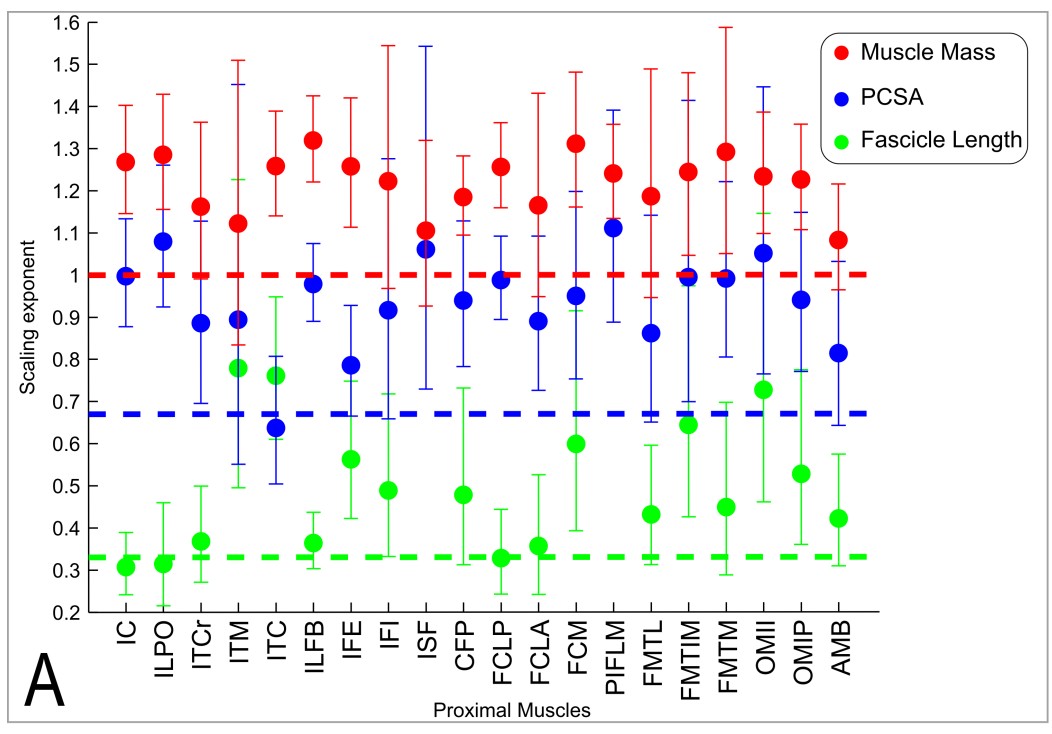

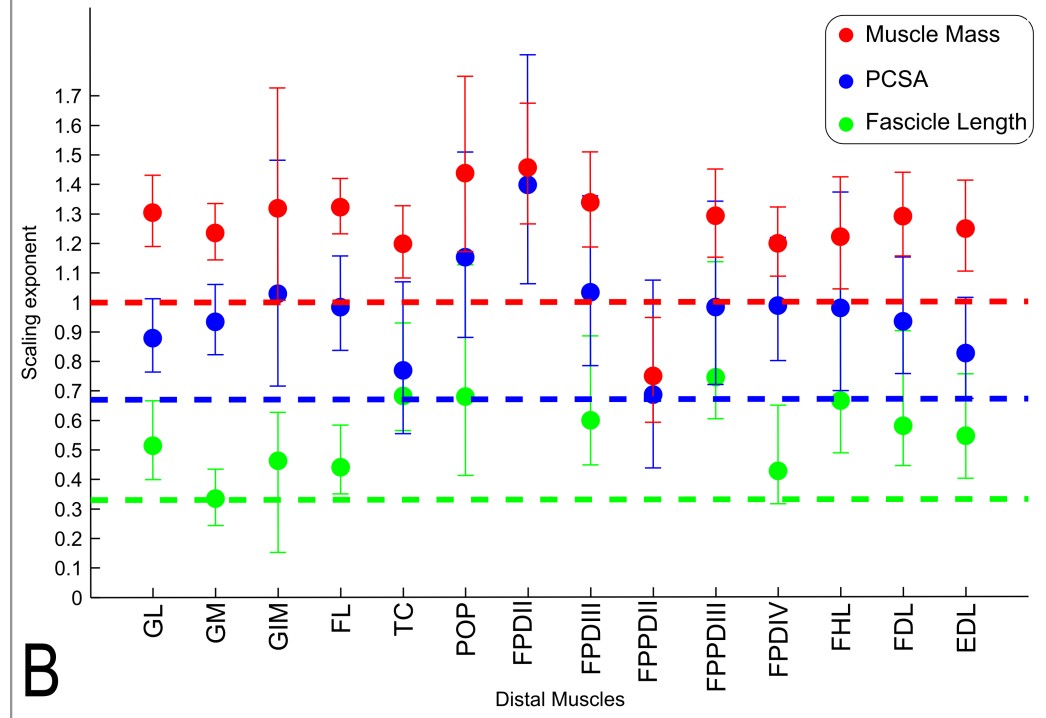

**Figure 5 Ontogenetic scaling exponents of muscle properties.** Ontogenetic scaling exponents and 95% confidence intervals (shown as error bars around mean exponent) for muscle mass (red), PCSA (blue) and fascicle length (green) for individual muscles in emu pelvic limbs. Abbreviations for muscles are in Table 1. Dashed lines indicate the expected isometric scaling exponent for each parameter. Data are for (A) proximal limb muscles and (B) distal limb muscles.

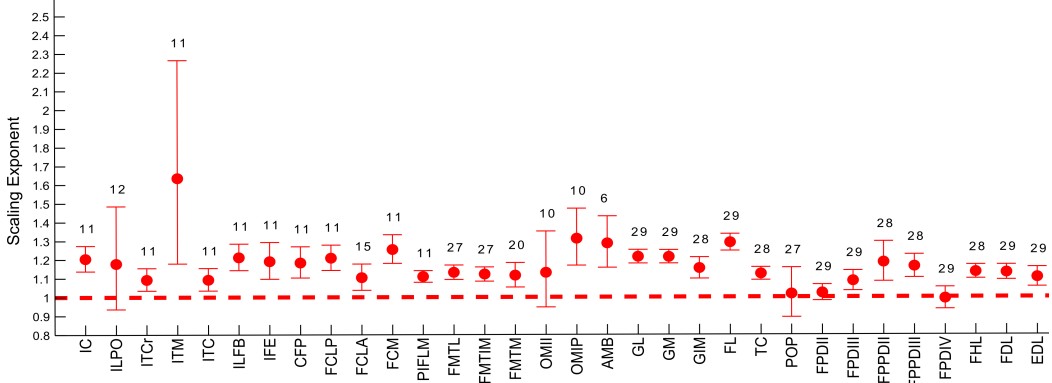

**Figure 6 Ontogenetic scaling exponents and 95% confidence intervals for masses of individual muscles in emu pelvic limbs, from the USA group.** Abbreviations for muscles are in Table 1. Dashed line indicates the expected isometric scaling exponent (1.0), and the number above each parameter indicates the number of muscles included in each regression analysis.

## Scaling of limb muscle masses

We found limb muscle mass as well as the masses of proximal (PLM) and distal limb muscles (DLM) to be tightly correlated with body mass across all three groups. The regression slope of limb muscle mass vs. BM was 1.16 ($1.05 < CI < 1.29$, $R^2 = 0.96$), whilst PLM had a value of 1.14 ($1.02 < CI < 1.27$, $R^2 = 0.96$) and DLM exhibited a slope of 1.20 ($1.09 < CI < 1.32$, $R^2 = 0.97$).

Consistent with the trends for the hindlimb more broadly, the individual muscles also generally showed positive ontogenetic allometry. Out of 34 muscles, 26 had slopes for $M_m$ vs. BM with their lower CI limit >1 (consistent with positive allometry), and only eight (ITCr, ITM, IFI, ISF, FCLA, FMTL, AMB and FPPDII) had a lower CI boundary for the regression slope lower than 1 (indicating potential negative allometry). Of the 26 muscles showing positive allometry of $M_m$, we found strong positive allometry (regression slopes with the lower boundary of the CI greater than 1.1) in 18/34.

Similarly, scaling patterns of the muscle masses for the USA group of emus (Fig. 6), showed similar scaling patterns to the UK group, with only five muscles having a lower CI boundary <1 (POP, ILPO, FPDIV, OBTII and FPDII) and the remaining having their CIs entirely within positive allometry values.

## Scaling of muscle fascicle length

In general, fascicle length ($L_f$) was only moderately well correlated with body mass due to substantial variation in the data (a combination of inevitable measurement errors, sampling bias and true biological variation), as usual for muscle fascicle measurements (e.g., *Allen et al., 2010*; *Allen et al., 2014*). The datasets for four muscles (ISF, PIFLM, FPDII and FPPDII) had a *p* value >0.05, so these are not presented (Table 3). Of the remaining 30 muscles, only 16/30 had $R^2$ values >0.5. Scaling of $L_f$ vs. BM showed a trend towards positive allometry for 18/30 muscles (lower limit of the slope's CI > 0.33), and for the remaining 12 muscles a slope of 0.33 was included in the CIs, so isometry could not be ruled out.

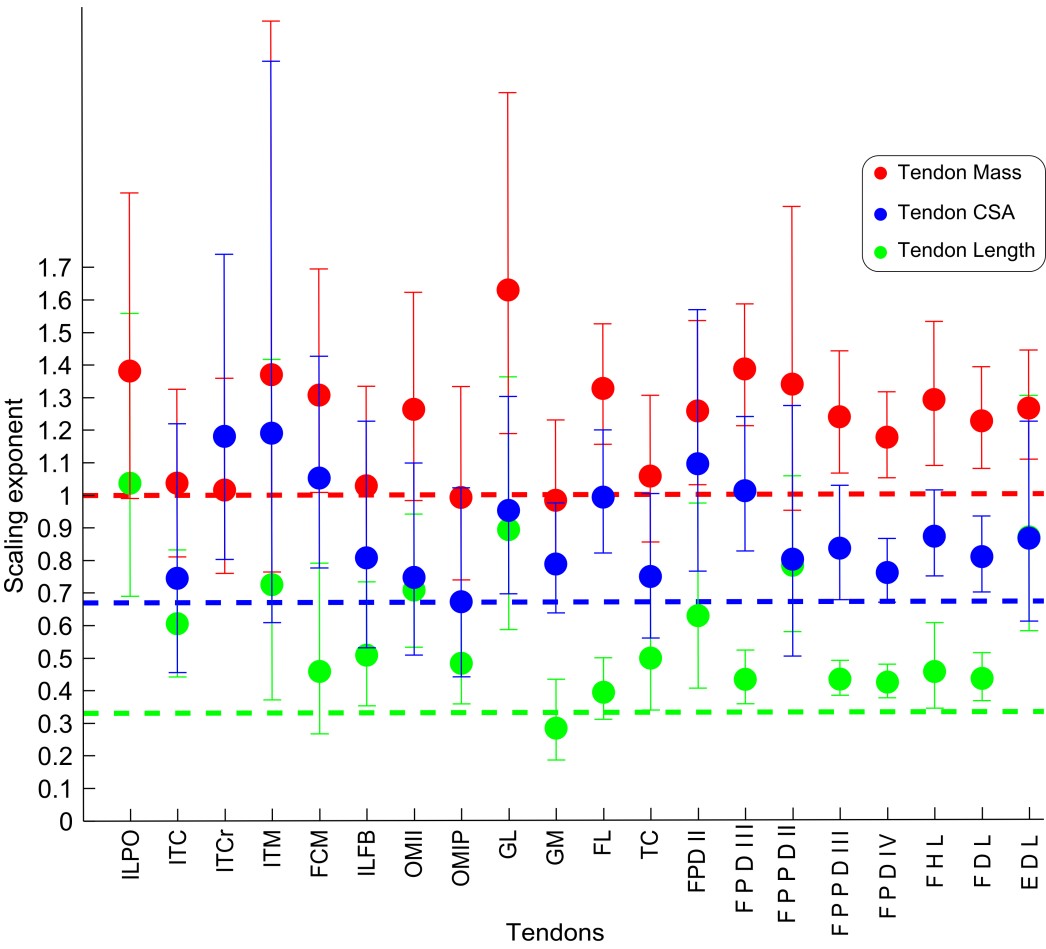

**Figure 7 Ontogenetic scaling exponents of tendon properties.** Ontogenetic scaling exponents and 95% confidence intervals for tendon mass (red), average cross-sectional area (blue) and length (green) for 20 individual muscles in emu pelvic limbs. Abbreviations for muscles are in Table 1. Dashed lines indicate the expected isometric scaling exponent for each parameter.

## Scaling of muscle PCSA

The lower boundary of the CIs of the scaling slope was greater than 0.67 (i.e., exhibiting positive allometry) for 27 muscles and a value <0.67 (suggesting a potential negative allometry of muscle PCSA in emus) was obtained for eight muscles (ITM, ITC, IFI, FMTL, AMB, TC and FPPDII) (Table 3).

## Scaling of tendon mass

We recorded tendon characteristics for 28 muscles (Table 4); the six muscles excluded did not have a discrete tendon at either of their attachments (CFP, FCLA, FCLP, IC, PIFLM, POP). We encountered difficulties in achieving a consistent method for tendon dissection and measurement of muscles with thin (IFE, AMB), very short (ISF and IFI) or multiple tendons (FMTM, FMTIM), which lead us to exclude data from these as well. The tendon of the GIM was included with the GM tendon, and the FMTL tendon was not measured because the muscle was transected at the proximal aspect of the large patellar tendon

**Table 3  Results of RMA linear regression of muscle architecture vs. body mass (BM) for the pelvic limb of *Dromaius novaehollandiae*, across ontogeny.**

| Muscle | $M_m$ vs BM | | | | | $L_f$ vs BM | | | | | PCSA vs BM | | | | |
|---|---|---|---|---|---|---|---|---|---|---|---|---|---|---|---|
| | Outliers | Slope | Lower 95% CI | Upper 95% CI | $R^2$ | Outliers | Slope | Lower 95% CI | Upper 95% CI | $R^2$ | Outliers | Slope | Lower 95% CI | Upper 95% CI | $R^2$ |
| AMB | 0 | 1.08 | 0.96 | 1.21 | 0.96 | 0 | 0.42 | 0.31 | 0.57 | 0.67 | 0 | 0.81 | 0.64 | 1.03 | 0.81 |
| CFP | 0 | 1.18 | 1.09 | 1.28 | 0.98 | 0 | 0.48 | 0.31 | 0.73 | 0.36 | 0 | 0.94 | 0.78 | 1.13 | 0.89 |
| EDL | 0 | 1.25 | 1.10 | 1.41 | 0.95 | 0 | 0.54 | 0.39 | 0.75 | 0.64 | 0 | 0.82 | 0.67 | 1.01 | 0.86 |
| FCLA | 1 | 1.16 | 0.95 | 1.43 | 0.87 | 1 | 0.36 | 0.24 | 0.53 | 0.51 | 1 | 0.89 | 0.73 | 1.09 | 0.87 |
| FCLP | 0 | 1.26 | 1.16 | 1.36 | 0.98 | 0 | 0.33 | 0.24 | 0.44 | 0.69 | 0 | 0.99 | 0.89 | 1.09 | 0.97 |
| FCM | 1 | 1.31 | 1.16 | 1.48 | 0.95 | 1 | 0.60 | 0.39 | 0.91 | 0.42 | 1 | 0.95 | 0.75 | 1.20 | 0.83 |
| FDL | 1 | 1.29 | 1.15 | 1.44 | 0.96 | 1 | 0.58 | 0.37 | 0.90 | 0.36 | 1 | 0.93 | 0.76 | 1.15 | 0.86 |
| FHL | 1 | 1.22 | 1.04 | 1.42 | 0.93 | 1 | 0.66 | 0.42 | 1.04 | 0.34 | 1 | 0.98 | 0.70 | 1.37 | 0.64 |
| FL | 0 | 1.32 | 1.23 | 1.42 | 0.98 | 0 | 0.44 | 0.33 | 0.58 | 0.73 | 0 | 0.98 | 0.84 | 1.16 | 0.91 |
| FMTIM | 0 | 1.24 | 1.05 | 1.48 | 0.90 | 0 | 0.64 | 0.43 | 0.97 | 0.40 | 0 | 0.99 | 0.70 | 1.41 | 0.57 |
| FMTL | 0 | 1.19 | 0.95 | 1.49 | 0.83 | 0 | 0.43 | 0.31 | 0.60 | 0.64 | 0 | 0.86 | 0.65 | 1.14 | 0.73 |
| FMTM | 0 | 1.29 | 1.05 | 1.59 | 0.86 | 0 | 0.45 | 0.29 | 0.70 | 0.31 | 0 | 0.99 | 0.80 | 1.22 | 0.85 |
| FPDII | 0 | 1.45 | 1.26 | 1.67 | 0.93 | 0 | – | – | – | – | 0 | 1.40 | 1.06 | 1.84 | 0.74 |
| FPDIII | 0 | 1.34 | 1.19 | 1.51 | 0.95 | 0 | 0.60 | 0.41 | 0.88 | 0.47 | 0 | 1.03 | 0.78 | 1.36 | 0.74 |
| FPDIV | 0 | 1.20 | 1.09 | 1.32 | 0.97 | 0 | 0.43 | 0.28 | 0.65 | 0.38 | 0 | 0.99 | 0.80 | 1.22 | 0.85 |
| FPPDII | 0 | 0.75 | 0.59 | 0.95 | 0.81 | 0 | 0.74 | 0.49 | 1.14 | 0.37 | 0 | 0.68 | 0.44 | 1.07 | 0.29 |
| FPPDIII | 0 | 1.29 | 1.15 | 1.45 | 0.96 | 0 | – | – | – | – | 0 | 0.98 | 0.72 | 1.34 | 0.67 |
| GIM | 0 | 1.32 | 1.01 | 1.73 | 0.75 | 0 | 0.46 | 0.34 | 0.63 | 0.69 | 0 | 1.03 | 0.72 | 1.48 | 0.54 |
| GL | 0 | 1.30 | 1.19 | 1.43 | 0.97 | 0 | 0.51 | 0.40 | 0.67 | 0.77 | 0 | 0.88 | 0.76 | 1.01 | 0.93 |
| GM | 0 | 1.24 | 1.14 | 1.33 | 0.98 | 0 | 0.34 | 0.26 | 0.43 | 0.77 | 0 | 0.93 | 0.82 | 1.06 | 0.95 |
| IC | 0 | 1.27 | 1.15 | 1.40 | 0.97 | 0 | 0.31 | 0.24 | 0.39 | 0.81 | 0 | 1.00 | 0.88 | 1.13 | 0.95 |
| IFE | 0 | 1.26 | 1.11 | 1.42 | 0.95 | 0 | 0.56 | 0.42 | 0.75 | 0.72 | 0 | 0.79 | 0.66 | 0.93 | 0.91 |
| IFI | 2 | 1.22 | 0.97 | 1.54 | 0.85 | 2 | 0.49 | 0.33 | 0.72 | 0.57 | 2 | 0.92 | 0.66 | 1.28 | 0.68 |
| IB | 0 | 1.32 | 1.22 | 1.42 | 0.98 | 0 | 0.36 | 0.30 | 0.44 | 0.89 | 0 | 0.98 | 0.89 | 1.07 | 0.97 |
| ILPO | 0 | 1.29 | 1.16 | 1.43 | 0.96 | 0 | 0.31 | 0.21 | 0.46 | 0.50 | 0 | 1.08 | 0.92 | 1.26 | 0.92 |
| ISF | 3 | 1.10 | 0.93 | 1.32 | 0.92 | – | – | – | – | – | 3 | 1.06 | 0.73 | 1.54 | 0.63 |
| ITC | 2 | 1.26 | 1.14 | 1.39 | 0.97 | 2 | 0.76 | 0.61 | 0.95 | 0.86 | 2 | 0.64 | 0.50 | 0.81 | 0.84 |
| ITCr | 0 | 1.16 | 0.99 | 1.36 | 0.92 | 0 | 0.37 | 0.27 | 0.50 | 0.68 | 0 | 0.89 | 0.70 | 1.13 | 0.80 |
| ITM | 2 | 1.12 | 0.83 | 1.51 | 0.75 | 2 | 0.78 | 0.49 | 1.23 | 0.39 | 2 | 0.89 | 0.55 | 1.45 | 0.29 |
| OMII | 0 | 1.23 | 1.10 | 1.39 | 0.95 | 0 | 0.73 | 0.46 | 1.15 | 0.27 | 0 | 1.05 | 0.76 | 1.45 | 0.65 |
| OMIP | 0 | 1.23 | 1.11 | 1.36 | 0.97 | 0 | 0.53 | 0.36 | 0.77 | 0.49 | 0 | 0.94 | 0.77 | 1.15 | 0.87 |
| PIFLM | 0 | 1.24 | 1.13 | 1.36 | 0.97 | – | – | – | – | – | 0 | 1.11 | 0.89 | 1.39 | 0.83 |
| POP | 2 | 1.44 | 1.17 | 1.76 | 0.88 | 2 | 0.68 | 0.41 | 1.13 | 0.22 | 2 | 1.15 | 0.88 | 1.51 | 0.79 |
| TC | 0 | 1.20 | 1.08 | 1.33 | 0.97 | 0 | 0.68 | 0.50 | 0.93 | 0.67 | 0 | 0.77 | 0.55 | 1.07 | 0.63 |

**Notes.**

$M_m$, muscle mass (kg); $L_f$, fascicle length (m); PCSA, physiological cross-sectional area ($m^2$).

**Table 4 Results of RMA linear regression of tendon dimensions vs. body mass (BM) for the pelvic limb of *Dromaius novaehollandiae*, across ontogeny.**

| Tendon | $M_{ten}$ vs BM | | | | | $L_{ten}$ vs BM | | | | | TCSA vs BM | | | | |
|---|---|---|---|---|---|---|---|---|---|---|---|---|---|---|---|
| | Outliers | Slope | Lower 95% CI | Upper 95% CI | $R^2$ | Outliers | Slope | Lower 95% CI | Upper 95% CI | $R^2$ | Outliers | Slope | Lower 95% CI | Upper 95% CI | $R^2$ |
| EDL | 0 | 1.26 | 1.10 | 1.44 | 0.94 | 1 | −0.81 | −1.07 | −0.61 | 0.75 | 0 | 0.86 | 0.61 | 1.22 | 0.58 |
| FCM | 0 | 1.31 | 1.01 | 1.69 | 0.86 | 0 | 0.46 | 0.27 | 0.79 | 0.34 | 0 | 1.05 | 0.78 | 1.43 | 0.81 |
| FDL | 1 | 1.22 | 1.08 | 1.39 | 0.95 | 1 | 0.43 | 0.36 | 0.51 | 0.91 | 1 | 0.81 | 0.70 | 0.93 | 0.94 |
| FHL | 1 | 1.29 | 1.09 | 1.53 | 0.91 | 1 | 0.45 | 0.34 | 0.60 | 0.74 | 1 | 0.87 | 0.75 | 1.01 | 0.93 |
| FL | 0 | 1.33 | 1.15 | 1.52 | 0.94 | 0 | 0.39 | 0.31 | 0.50 | 0.81 | 0 | 0.99 | 0.82 | 1.20 | 0.88 |
| FPDII | 0 | 1.26 | 1.03 | 1.53 | 0.87 | 0 | 0.63 | 0.40 | 0.97 | 0.32 | 0 | 1.09 | 0.76 | 1.57 | 0.56 |
| FPDIII | 0 | 1.38 | 1.21 | 1.58 | 0.94 | 0 | 0.43 | 0.36 | 0.52 | 0.88 | 0 | 1.01 | 0.82 | 1.24 | 0.86 |
| FPDIV | 0 | 1.17 | 1.05 | 1.31 | 0.96 | 0 | 0.42 | 0.37 | 0.48 | 0.95 | 0 | 0.76 | 0.67 | 0.86 | 0.95 |
| FPPDII | 0 | 1.34 | 0.95 | 1.88 | 0.60 | 0 | 0.78 | 0.58 | 1.06 | 0.69 | 0 | 0.80 | 0.50 | 1.27 | 0.24 |
| FPPDIII | 0 | 1.24 | 1.06 | 1.44 | 0.92 | 0 | 0.43 | 0.38 | 0.49 | 0.95 | 0 | 0.83 | 0.68 | 1.03 | 0.85 |
| GL | 0 | 1.63 | 1.19 | 2.23 | 0.66 | 0 | 0.89 | 0.59 | 1.36 | 0.38 | 0 | 0.95 | 0.69 | 1.30 | 0.66 |
| GM | 0 | 0.98 | 0.78 | 1.23 | 0.83 | 0 | 0.28 | 0.18 | 0.43 | 0.37 | 0 | 0.79 | 0.64 | 0.97 | 0.85 |
| IB | 1 | 1.03 | 0.79 | 1.33 | 0.79 | 1 | 0.51 | 0.35 | 0.73 | 0.57 | 1 | 0.81 | 0.53 | 1.23 | 0.43 |
| ILPO | 2 | 1.38 | 0.99 | 1.93 | 0.68 | 2 | 1.04 | 0.69 | 1.56 | 0.51 | – | – | – | – | – |
| ITC | 3 | 1.04 | 0.81 | 1.33 | 0.84 | 3 | 0.61 | 0.44 | 0.83 | 0.74 | 3 | 0.75 | 0.46 | 1.22 | 0.34 |
| ITCr | 1 | 1.02 | 0.76 | 1.36 | 0.73 | – | – | – | – | – | 1 | 1.18 | 0.80 | 1.74 | 0.52 |
| ITM | 7 | 1.37 | 0.76 | 2.46 | 0.43 | 6 | 0.72 | 0.37 | 1.42 | 0.09 | 7 | 1.19 | 0.61 | 2.33 | 0.21 |
| OMII | 0 | 1.26 | 0.98 | 1.62 | 0.79 | 0 | 0.71 | 0.53 | 0.94 | 0.72 | 0 | 0.75 | 0.51 | 1.10 | 0.48 |
| OMIP | 0 | 0.99 | 0.74 | 1.33 | 0.70 | 1 | 0.48 | 0.36 | 0.65 | 0.71 | 1 | 0.67 | 0.44 | 1.02 | 0.43 |
| TC | 0 | 1.06 | 0.85 | 1.30 | 0.85 | 0 | 0.50 | 0.34 | 0.73 | 0.47 | 0 | 0.75 | 0.56 | 1.00 | 0.71 |

**Notes.**

$M_{ten}$, tendon mass (kg); $L_{ten}$, tendon length (m); TCSA, tendon cross-sectional area ($m^2$).

for studies of patellar tendon morphology by *Regnault, Pitsillides & Hutchinson (2014)*. Thus data are presented for the tendons of 20 muscles. The major gastrocnemius tendon resulting from the fusion of the tendons of the three gastrocnemius muscles was dissected by transecting the tendon of the GL at the site of fusion onto the common tendon; therefore the GM remained with the extensive common portion of the tendon, which distally was transected at its insertion onto the fibrous scutum at the level of the ankle joint.

The scaling slopes for tendon mass indicate positive allometry in 10 out of 20 tendons (lower CI boundary >1) across emu ontogeny. The masses for the remaining ten tendons scaled with isometry (lower CI < 1.0, upper CI > 1.2).

### Scaling of tendon length

We measured $L_{ten}$ for the same 20 muscles for which we obtained tendon masses (Table 4), from the end of the muscle belly to the insertion. Statistical analysis of one muscle (ITCr) led to exclusion of this muscle because the *p* value was >0.05. For the other 19 tendons, the general scaling trend was towards strong positive allometry, with 16 muscles having the lower limit of the CI >0.33. In three muscles (FCM, GM, FL), the lower CI for tendon length was <0.33, indicating isometry for length in these tendons. Given these patterns, we infer a general trend for positive allometry of tendon length in growing emus.

### Scaling of tendon cross-sectional area

Average TCSA was calculated for the same 20 tendons as above (Table 4). The dataset for ILPO had a *p* value >0.05 and was excluded. Of the 19 remaining tendons, 10 showed a lower CI limit of the slope consistent with positive allometry (>0.67). The remaining nine tendons showed ontogenetic isometry for TCSA.

## DISCUSSION

Emus, like other ratites and other precocial birds, must have locomotor independence from hatching and develop into large, running adult birds within 16–18 months (*Davies & Bamford, 2002*). Taking into consideration their initial development within the egg, their ontogeny poses interesting questions about their locomotor development, related to our study's aims, such as: How do muscle structure and anatomy change to accommodate precocial development in emus? What are the strategies that growing emus use to maintain tissue mechanical safety factors during rapid development of cursorial morphology and high-speed locomotor abilities? Our data suggest some answers to these questions, as follows.

### Scaling patterns across ontogeny

We found positive allometry of emu pelvic limb muscle masses, indicating that most muscles get become significantly more powerful (in relative and absolute terms) as the animals grow. However, the functional relevance of this observation is slightly mitigated by a less marked positive allometry of PCSA (and therefore maximal muscle force), driven by a trend for fascicle length that is closer to isometry (i.e., preserving geometric similarity).

In the proximal part of the pelvic limb of emus, the developmental and functional mechanics appear to rely on the arrangement of large and metabolically expensive muscles (ILPO, ILFB, IC, FCLP and FMTL) to provide the wide range of motion of the knee joint (and hip, during faster running) in combination with a relatively short femur that scales close to isometry. This arrangement also leads to a proximal to distal gradient of muscle mass, which has been previously reported for other birds (*Paxton et al., 2010*; *Smith et al., 2006*) and has long been thought to favour energy-savings by keeping the distal end of the limb light and its muscles dependent on springy tendons. The proximal-distal gradient also concentrates large, power-generating muscles in the proximal limb (*Alexander, 1974*; *Alexander, 1991*) with large moment arms (*Hutchinson et al., 2014*; *Smith et al., 2007*) and thus the ability to produce the considerable joint moments needed for high-speed running (*Hutchinson, 2004a*; *Hutchinson, 2004b*).

The distal limb, on the other hand, is heavily dependent on the triad of M. gastrocnemius (GL, GIM and GM) along with: M. fibularis longus (FL); both ankle extensors; as well as M. tibialis cranialis (TC); M. extensor digitorum longus (EDL); and both ankle flexors. Combined, these muscles constitute 80% of the muscle mass and 60% of the force-generating capacity (PCSA) of this portion of the limb. The unusual proportion of body mass taken up by the ankle extensors has been noted before (*Hutchinson, 2004a*) and is likely an ancestral characteristic of birds (e.g., *Paxton et al., 2010*) but is taken to an extreme in large ratites (e.g., *Smith et al., 2006*).

Further distally, the long and slender tarsometatarsus bone lends itself well as a support for the long tendons of the digital flexor muscles which in turn provide essential springs used in support and propulsion of the limbs and body. The relatively small muscles and long tendons of the digital flexors make them likely to operate mainly as energy storage devices at faster speeds, as seen in other species like horses and smaller running birds (*Biewener, 1998*; *Daley & Biewener, 2011*). The positive allometry of many tendon properties in emus is in line with this increase in force-generating capacity seen during ontogeny. As in most other birds, the tendons running along the tarsometatarsus are almost exclusively on the cranial and caudal (dorsal/plantar) side. It would also be interesting to know the effect on bone strains from this "bow and arrow" anatomical arrangement between the tarsometatarsus and the dorsal/plantar tendons to see if it influences the predominantly torsional loads experienced by the two proximal pelvic limb bones (*Main & Biewener, 2007*).

For these spring-like tendons, a trade-off between muscle force and tendon elasticity does not seem to occur in emus. This lack of a trade-off is indicated by the similar scaling patterns of the cross-sectional areas of the digital flexor muscles and tendons, both of which trend towards positive allometry across emu ontogeny. As seen in other species (*Ker, Alexander & Bennet, 1988*), the relative increases in the cross-sectional areas of tendons might maintain tendon safety factors (maximal stresses before failure vs. *in vivo* maximal stress) as emus increase in size. However, tendons might also change their biomechanical properties (Young's modulus) with age, as seen in other species (*Shadwick, 1990*; *Thorpe et al., 2014*), therefore influencing biomechanical interpretations of the data presented here.

**Peer**J

Without measuring tendon elastic modulus with age, it is difficult to interpret how tendon stiffness and safety factor might change with age in emus.

To complement data from a prior study showing the scaling patterns of the cross-sectional areas of the femur and tibiotarsus of emus to be close to isometry (*Main & Biewener, 2007*), here we analysed the scaling patterns of the lengths of the three longest limb bones and the first phalanx of the third toe. Our data indicate positive allometry of the two longer bones, the tibiotarsus (lower CI limit = 0.37) and tarsometatarsus (lower CI limit = 0.39), but a less marked positively allometric scaling trend for the femur (lower limit of CI = 0.34) and for the first phalanx of digit III (lower CI limit = 0.33). These results differ from those reported for another ratite, the greater rhea (*Rhea americana*), in which only the tarsometatarsus showed positive allometry (*Picasso, 2012*) but interestingly are in line with general interspecific scaling exponents found for pelvic limb bone lengths across different species of palaeognaths (*Cubo & Casinos, 1996*). Considering our results, if similar cross-sectional geometry is assumed along the length of the bone shafts, this would lead to an increase in strains (at least in bending, due to larger moments) at the mid-shaft with increasing body mass. However, changes in cross-sectional areal geometry have been shown to lead to slight positive allometry of the cross-sectional geometry of avian limb bones across species (*Doube et al., 2012*) and ontogenetically (*Main & Biewener, 2007*). As these geometrical changes might not suffice to explain the increases in strain magnitudes seen during ontogeny, they leave unexplained the role of internal forces (soft tissues) on bone mechanics and consequently their influence on bone morphology during growth.

Although there are very limited data on the ontogeny of skeletal muscle physiology, experiments in mice and cats (*Close, 1964*; *Close & Hoh, 1967*) demonstrate that although muscle force: velocity parameters change from newborns to adults, these changes appear to occur in a relatively short period and therefore newborn muscle, after the first few days of life, becomes similar to that of adults. However, mice and cats, like many other mammals, are born with neuromotor immaturity (*Muir, 2000*), in contrast to emus. It is therefore reasonable to speculate that, like other birds (*Gaunt & Gans, 1990*), emus are unlikely to have appreciable changes in muscle physiology during growth. Thus changes in functional (e.g., maximal force-generating capacity) and biomechanical parameters should be detectable by anatomical studies such as ours.

Few studies have quantified the ontogenetic scaling patterns of limb musculature in birds (*Carrier & Leon, 1990*; *Dial & Carrier, 2012*; *Paxton et al., 2014*; *Picasso et al., 2012*; *Picasso, 2014*), but positive allometry predominates in the muscle masses involved in the major adult mode of locomotion (flying vs. cursorial). In the Californian gull, the M. gastrocnemius scaled isometrically but the M. pectoralis had strong positive allometry with an inflection point when the fledglings started exercising their wings (*Carrier & Leon, 1990*). *Paxton et al. (2014)* and *Paxton et al. (2010)* both recently reported the ontogenetic scaling patterns of the musculature of a highly modified galliform, the broiler chicken. These birds, unsurprisingly due to their selective breeding, were found to have positive allometry of muscle masses of the main pelvic limb muscles but isometry of the fascicle lengths (*Paxton et al., 2014*), a pattern that is nonetheless similar to our findings.

*Picasso et al. (2012)* found quite similar scaling patterns across rhea ontogeny: an average 64-fold increase in pelvic limb muscle mass from 1 month of age to adulthood whilst only a 34-fold increase in body mass. In a later study, where scaling exponents were calculated, a more generalised positively allometric scaling was found in these South American ratites compared to emus: with all muscle masses but two (where isometry was evident) scaling with positive allometry (slopes ~1.3). Total limb muscle mass of rheas scaled with an exponent of 1.18 (*Picasso, 2014*), which is similar to our value of 1.16. Together, these data suggest that positive allometry prevails across ontogeny for leg muscles in extant birds with precocial development; potentially a homologous feature of their development that is quite unlike the isometry prevailing in their closest extant relatives, Crocodylia (*Allen et al., 2010*; *Allen et al., 2014*).

*Dial & Carrier (2012)* suggested that birds must optimise their energy consumption to achieve their ultimate functional gait whilst channelling resources to their precocial gait (*Dial & Carrier, 2012*) (running vs. swimming or flying). Ratites are unusual for birds in that they solely have terrestrial gaits throughout their life and, in the case of emus, their wings have atrophied to such an extent that they should not present much metabolic competition to hindlimb development. Considering the approximately isometric overall scaling of kinematic parameters (e.g., stride lengths, stride frequencies, duty factors) seen in ratites (*Main & Biewener, 2007*; *Smith, Jespers & Wilson, 2010*), it is likely that this increase in muscle masses will lead to a limb that is adapted for power production and perhaps (considering our less allometric tendon results) elastic energy storage/return. The former is also supported by metabolic studies which found a predominance of fast fibres in the M. gastrocnemius of adult emus (*Patak, 1993*), although more studies of muscle physiology in emus and other ratites would be valuable.

The need for locomotor independence and high performance in vulnerable, young, precocial and cursorial birds might favour allometry of muscle architecture (*Carrier, 1996*). If so, could adult muscle phenotypes be a reflection of the locomotor needs during early development and therefore be overdesigned for their demands? Alternatively, negative allometric scaling in the musculoskeletal system may occur as seen in goats (*Main & Biewener, 2004*) and jackrabbits (*Carrier, 1983*). It is hard to draw an inference from our data, because the overall positive allometry seen in the pelvic limb musculature could indicate a necessity to grow faster and stronger to adulthood to compensate for a juvenile disadvantage, or could reflect selective pressures on the locomotor ontogeny of emus in which muscles congenitally primed for fast growth during adolescence could lead to continued growth past an optimum in adulthood. Although direct measurements of maximal performance of complex locomotor systems is problematic, a modelling approach using the data presented here could be a valid approach to answer this question.

## How well are farmed emus representative of the species overall?

Although emu farming is relatively common, its goal is to extract meat, oil and skin and therefore these birds are not bred in captivity for their locomotor behaviour, nor do they suffer strong predatory pressures on it. The diet of captive bred birds as well as their relative

sedentary regime when compared to wild animals is likely to influence tissue development and distribution. However, as farming of these birds is a recent activity and it is not a highly specialised or intense process as with other domesticated species (*Goonewardene et al., 2003*), it is unlikely that heritable traits of the emu musculoskeletal system have been significantly altered. Therefore, we expect the muscle distribution and scaling patterns of our emus to be similar to wild emus.

By presenting muscle mass data from two distinct groups of birds (UK and USA groups), we established that these groups at least have similar scaling patterns, ruling out any potential bias imposed by different breeding regimes. With regards to diet, it was apparent that our birds were carrying a significant amount of subcutaneous and peritoneal fat; likely encouraged by their *ad libitum* access to a commercial pelleted diet. The influence of body fat on our scaling results is hard to test with the available data, but *Hutchinson et al. (2014)* noted a possible reduction in relative muscle masses in wild vs. captive bred ostriches, which could also apply to emus. Regardless, it is less certain that the scaling patterns for muscle/tendon architecture observed here would differ in wild vs. captive emus.

## CONCLUSIONS

We have provided a new dataset on the ontogenetic scaling of pelvic limb anatomy and muscle architectural properties of a cursorial bird (the first complete architectural dataset of its kind), and we have done this using a group of 17 emus across a tenfold increase in body mass. A marked trend of positive allometry of muscle masses and PCSAs is accompanied by less marked positive allometry of fascicle lengths. Tendons, especially the long digital flexors, also demonstrate positive allometry of their lengths, as do the two longer limb bones (tibiotarsus and tarsometatarsus). We have illuminated the ontogenetic adaptation of the musculoskeletal system in an extreme example of size variation during rapid growth. Our dissections refined the myology of the pelvic limb in emus (Table 1 and Figs. 1–3) and found some anatomical aspects that were previously misunderstood. This is important as functional studies depending on inaccurate anatomical accounts of the myology could obtain unrealistic results from biomechanical models using such data (*Goetz et al., 2008*; *Hutchinson et al., 2014*). This work should be a valuable resource for future functional, comparative and evolutionary studies of emus, other birds and extinct related animals.

## ACKNOWLEDGEMENTS

We thank Jack Machale, Emily Sparkes, Kyle Chadwick, Charlotte Cullingford, Sophie Regnault and Chris Basu who helped with the dissections. Craig McGowan provided assistance with muscle dissections in the USA emu sample. A special thank you goes to Vivian Allen for providing the custom designed R code that we used to perform the regression analysis, as well as valuable intellectual discussions. We also thank Ashley Heers and Diego Sustaita as well as an anonymous reviewer and reviewer Trevor Worthy for their helpful comments on an earlier draft of this manuscript.

### Funding

Our funding bodies are the Fundação para a Ciência e Tecnologia-FCT (Portuguese Government-Foundation for Science and Technology) for PhD studentship funding for LPL (Grant Code SFRH/BD/74439/2010), the Royal Veterinary College, and grant number BB/I02204X/1 from the British Biotechnology and Biological Sciences Research Council. The funders had no role in study design, data collection and analysis, decision to publish, or preparation of the manuscript.

### Grant Disclosures

The following grant information was disclosed by the authors:
Fundação para a Ciência e Tecnologia-FCT (Portuguese Government-Foundation for Science and Technology).
LPL: SFRH/BD/74439/2010.
Royal Veterinary College: BB/I02204X/1.
British Biotechnology and Biological Sciences Research Council.

### Competing Interests

John R. Hutchinson is an Academic Editor for PeerJ.

### Author Contributions

- Luis P. Lamas, Russell P. Main and John R. Hutchinson conceived and designed the experiments, performed the experiments, analyzed the data, contributed reagents/materials/analysis tools, wrote the paper, prepared figures and/or tables, reviewed drafts of the paper.

### Animal Ethics

The following information was supplied relating to ethical approvals (i.e., approving body and any reference numbers):

Birds were obtained from our ongoing research examining emu ontogenetic biomechanics (conducted with ethical approval under a UK Home Office license: PPL 70 7122).

### Supplemental Information

Supplemental information for this article can be found online at http://dx.doi.org/10.7717/peerj.716#supplemental-information.

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
