# Peer review of "Ontogenetic scaling patterns and functional anatomy of the pelvic limb musculature in emus (Dromaius novaehollandiae)"

_PeerJ, doi:10.7717/peerj.716_

## Round 0.1 · original submission · Major Revisions

I have received two reviews of your paper. Both reviews found the paper a worthwhile contribution. Review 1 made minor editorial comments directly on the manuscript and overall liked the paper. However, reviewer 2 felt that the overall writing of the paper could be improved and the manuscript shortened. This review also had concerns about data analysis and redundancy between text and tables. As a result, I may need to send out your revised manuscript for additional review. Please take care to document your changes and responses to the reviewers extensively.

Reviewer 1 ·

Basic reporting

The basic reporting of the information in this paper is not acceptable for publication. In general it is poorly written and the entire paper can be made much more concise. The results section is a tedious description of the all of the information contained in the tables, which is not necessary. For example, it is not necessary to discuss the range of confidence intervals for each parameter - simply state whether or not there was substantial variation. Throughout the manuscript, several sentence are awkward and confusing and many statements in the Discussion are not supported. A more meaningful interpretation of the findings in the Discussion would also greatly improve the impact of this study. Overall, I do believe this manuscript can reach publication quality writing with the input of the senior authors on the paper.

Experimental design

The experimental design of the study is sound and leverages work done by two groups. I believe this is a strength of the paper.

Validity of the findings

There is one fundamental flaw in the data as reported that must be addressed before the manuscript can be accepted. Specifically, in is inappropriate to normalize the data presented in figure 4 and discussed on lines 245 - 253 (page numbers would be helpful in the future) by isometric values. Given the substantial allometric scaling patterns reported in this study, normalizing parameters such as muscle mass by body mass heavily skews the data for towards larger animals. This normalization adds little to the study and should be removed completely.

Additional comments

Overall, this is a data rich study that I feel will be valuable to the scientific community. However, it must be substantially revised before it is acceptable for publication. The entire manuscript can be made much more concise and I feel that the paper would benefit greatly from a more organized and in depth interpretation of the finding.

·

Basic reporting

OK

Experimental design

OK

Validity of the findings

OK

Additional comments

Please see attached file for suggestions

---

## Round 0.2 · accepted · Accept

After reviewing the rebuttal letter and the revised manuscript, I feel you have adequately addressed the reviewers comments, and I am pleased to recommend acceptance of your article to PeerJ.